# Estimation of Technical, Allocative, and Economic Efficiencies for Smallholder Broiler Producers in South Africa

**Lindikaya W. Myeki** [1,2,*], **Nkhanedzeni B. Nengovhela** [3,4], **Livhuwani Mudau** [3], **Elvis Nakana** [2] **and Simphiwe Ngqangweni** [2]

1   Department of Agricultural Economics, University of the Free State, Bloemfontein 9300, South Africa
2   National Agricultural Marketing Council, MERC Division, 536 Francis Baard St, Arcadia, Pretoria 0002, South Africa
3   Department of Agriculture Land Reform and Rural Development, Pretoria 0001, South Africa
4   Department of Agriculture and Animal Health, University of South Africa, Florida 1710, South Africa
*   Correspondence: lindikayam@yahoo.co.za or lindikaya@namc.co.za

**Abstract:** The prevailing economic conditions as a result of COVID-19, climate change, and the Russia–Ukraine conflict, have led to renewed global interest in the efficiency of the agricultural sector. As a result of this, we investigated the efficiencies of 64 broiler producers in three districts covering the North West and Limpopo provinces in South Africa from 2019 to 2022, using a two-stage data envelopment analysis method with input orientation. The results show that producers operate on the upper bounds toward efficiency, but room for improvement still exists at 10%, 20%, and 28% on technical, allocative, and cost efficiencies. This indicates that inputs can still be reduced without changing the level of output and that the input combination is incompatible with cost minimization. Consequently, only 13%, 8%, and 4% of the sampled broiler producers exhibited technical, allocative, and cost efficiencies, respectively; the majority were women. The Vhembe, Capricorn, and Dr. Kenneth Kaunda (DRKK) districts had vastly different scores for each efficiency type, indicating that their differences in resource endowments, technology, and climate, necessitate the formulation of district-specific policies. The mortality rate, heating costs, and investments in health emerged as significant efficiency determinants. Overall, the findings provide insights into refocusing the country's poultry sector in light of current shocks and the notable aspirations of the poultry master plan.

**Keywords:** efficiency broiler; producer; DEA; South Africa

## 1. Introduction

Broilers are birds that are bred and grown for meat production purposes. They command 76% of the total bird population (in the poultry sector) in South Africa, with an estimated gross value of ZAR 47.96 billion and total direct employment of 49,887 people [1]. Broiler meat is the most widely consumed meat product in the world because it is relatively inexpensive with very few or no consumption barriers [2]. Thus, broiler meat production is expected to rise as it is (globally) a fast-growing industry, and due to the increase in per capita consumption in third world countries [2]. South Africa is not exempt from this; currently, domestic production remains insufficient to meet the growing local consumption [3]. For instance, during 2018/2019, South Africa produced a total of 1.76 million tons of broilers but consumed 2.3 million tons that same year. The gap between consumption and production continues to widen; the country became a net importer of broiler meat during the period under review. The per capita consumption of broiler meat in South Africa has increased from 39.19 kg per person in 2017/2018 to 39.85 kg per person in 2018/2019, an approximate 1.7% increase [3]. This necessitates an increased focus on enhancing the productivity and efficiency of the domestic broiler industry with special attention to small-holder broiler producers (because they possess great potential).

In South Africa, broiler production is conducted in all nine provinces by commercial producers and smallholders [4]. Commercial broiler producers are few, but are fully integrated into the mainstream economy [3] with well-organized marketing and distribution channels that involve a few large supermarket chains, and quick-service (fast-food) restaurants, including international franchises [5,6]. They produce over 75% of broiler meat in the country [1]. Smallholder broiler producers have production operations involving either contract farming or non-contract selling solely to the informal market. The actual number of these producers remains unknown due to the lack of a national database. However, the authors of [5] claim that smallholder broiler producers operate at very low capacities that range from 100 to 5000 birds per cycle. A recent study conducted by [4] covering eight provinces shows that smallholder farmers place 1300 birds per cycle and sell approximately 1100, generating between ZAR 15,000 and ZAR 25,000 of net farm income. According to [7], the only reason they are still able to compete is that they rely on niche (live) markets, where the prices of live birds are high, compensating for the low numbers and the period it takes to sell all stock. Despite receiving enormous support from the government, their efficiencies in the production of broilers remain unbounded, and the sources of these efficiencies remain inconclusive.

Broiler farming efficiency has been well researched throughout the world, using two main frontier methodologies (non-parametric data envelopment analysis and parametric stochastic frontier analysis) that have unique advantages and drawbacks. Although some research findings report greater levels of technical efficiency [8–13], a large number of studies have demonstrated the scopes needed for improvements. For instance, reference [14] assessed the technical efficiencies of 60 broiler producers in Pakistan and found a 20% scope for improvement in broiler production. Using the same sample size, reference [15] found technical inefficiency of 25% in broiler farmers in Nigeria. From a sample of 52 broiler producers, reference [16] estimated the technical, allocative, and economic efficiencies and found 37%, 38%, and 62% potential for improvement. The findings of [17] revealed the potential to improve technical efficiency by 28% in the Chinese broiler sector. Most of these studies used cross-sectional data to only provide the estimates of technical efficiency. Agriculture efficiency studies in South Africa have been conducted on cattle production [18,19], maize [20], sheep production [21], mixed farming [22], and dairy [23] using different frontier methodologies and datasets. However, [24] is the only study in the country that has attempted to analyze broiler efficiency. The study was conducted in Limpopo province using cross-sectional data obtained from 86 broiler producers and applying a stochastic frontier analysis with Cobb–Douglass functional form. The results show that efficiency is explained by feed, vaccines, and stock sizes, but affected by the level of education, farming experience, age, and gender. The study was silent on the distribution of efficiency scores and only focused on a relatively small part of South Africa. In addition, the study did not provide estimates of allocative and cost efficiencies. Therefore, this paper seeks to fill this research gap.

This paper has two research questions—what are the technical, allocative, and cost efficiencies of smallholder broiler producers in three districts covering the North West and Limpopo provinces? What are the determinants of the three efficiency types? To address the first question, a data envelopment analysis (DEA) method with input orientation was applied to pooled data obtained from 64 smallholder producers to derive estimates for the three efficiency types. We employed the fractional regression model (FRM) to answer the second research question. In this way, the study uniquely contributes in three ways. First, it expands the current understanding of the efficiency of smallholder broiler producers in South Africa by exploring three efficiency types as opposed to a single efficiency by the previous research. Second, it provides an indirect evaluation of government support for smallholder broiler farmers. Third, we departed from the tradition of using the Tobit regression model to assess the determinants to utilize FRM. Fourthly, the findings provide insights into achieving the goals of the poultry master plan in 2023. The remainder of the paper is as follows: Section 2 presents the methodical approach for the study with special

emphasis on data and the estimation procedure; Section 3 presents the findings of the study. Section 4 provides the discussion of the results and Section 5 concludes the paper. Throughout the study, technical efficiency (TE) refers to the ability of broiler producers to minimize inputs without changing the output; allocative efficiency (AE) refers to the ability of a firm to use inputs in optimal proportions, given their respective prices and production technology.

## 2. Materials and Methods

### 2.1. Study Sites and Data Information

The study was conducted in North West and Limpopo, South Africa. As can be seen from Figure 1, the former covers four district municipalities (37 to 40); whereas the latter consists of 5 districts (33 to 36 and 47). However, the study focused on three specific districts. These districts included Dr Kenneth Kaunda (DRKK), denoted by 40 in Figure 1, Vhembe (34), and Capricorn (35). The samples from the 3 districts constituted of 64 smallholder broiler producers using the purposive sampling method. These producers participated in a production scheme supported by the Department of Agriculture Land Reform and Rural Development (DALRRD), to equip smallholder broiler producers with the skills and knowledge necessary to operate their farms successfully and sustainably. In its implementation, the scheme involved efforts to reduce mortality rates, increase capacity through training and education, and improve productivity and market access. This led to the collection of farm business performance data, which included information on the production level, product price, employment creation, and other parameters on the beneficiaries. While slight improvements were observed from these farmers, their efficiency levels in broiler production remained unknown. Therefore, our study filled this knowledge gap.

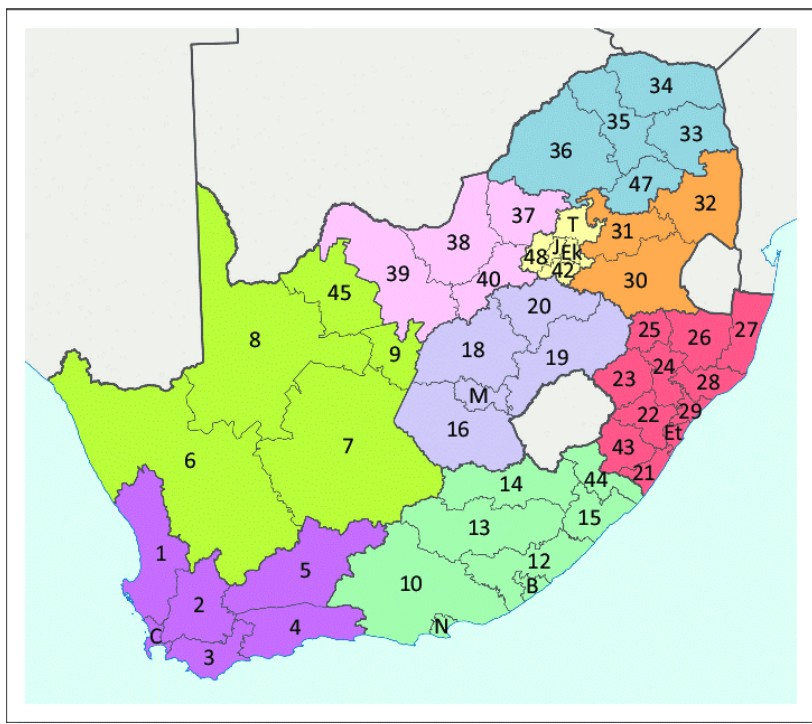

**Figure 1.** The distribution of study sites by provinces in South Africa. Source: Brinley. The area in pink colour covering 37–40 is North West province. The area covered by numbers 33–36 and 47 is Limpopo province. The number represent the district municipalities. DRKK, denoted by 40, Capricorn is 35 and Vhembe is 34.

We used one output variable, defined as the number of live broiler birds sold. The input variables included: (i) feed, which was derived by averaging the three types of feed,

the starter, grower, and finisher. This variable was measured in terms of 50 kg bags. (ii) Stock, defined as the number of birds or day-old chicks reared, and (iii) bedding, which refers to the wood shaving material used on the floors of broiler houses. These variables were matched by their respective prices for the purpose of estimating allocative and economic efficiencies. The determinants for the efficiency types were assessed using variables: mortality rate, measure in percentages, heating costs (e.g., the electricity expenses), a dummy for the gender (where 1 is male, 0 otherwise), and a dummy for the district (where 1 is the Capricorn district and 0 otherwise). The overall variable choices were influenced by previous research studies [11,24,25]. As can be seen from Table 1, there were clear differences between the outputs and inputs. For instance, the output grew at an average of 398 sales, ranging from 88 to 2016 birds. This was achieved by using an average of 15 bags of 50 kg feed with a range from 3 to 80, an average stock of 511 chicks, and bedding at 16 kg.

**Table 1.** Descriptive summary of the output and input variables.

| Variable | Mean | Std | Minimum | Maximum |
|---|---|---|---|---|
| Output | | | | |
| Sales (n) | 397 | 420 | 88 | 2016 |
| Inputs | | | | |
| Feed (50 kg bag) | 15 | 17 | 3 | 80 |
| Bird stock (n) | 511 | 678 | 100 | 4104 |
| Bedding (kg) | 16 | 20 | 0 | 82 |
| Prices | | | | |
| Sales (R/bird) | 397 | 420 | 88 | 2016 |
| Feed (R/bag) | 324.60 | 47.72 | 247.00 | 605.00 |
| Stock (R/chick) | 8.37 | 0.99 | 4.25 | 9.50 |
| Bed (R/kg) | 30.80 | 7.54 | 12.50 | 75.00 |
| Determinants | | | | |
| Mortality (%) | 9.35 | 8.50 | 0.00 | 43.42 |
| Heating cost (R) | 1188.38 | 2595.46 | 50.00 | 14,250.00 |
| Dummy gender | 0.65 | 0.48 | 0.00 | 1.00 |
| Dummy district | 0.48 | 0.50 | 0.00 | 1.00 |

*2.2. Estimation Procedure*

Different researchers have estimated the efficiencies using two prominent frontier methodologies [26]. One method involves regression-based SFA, which requires the specification of the frontier's functional form, the satisfaction of normality assumptions about the sample, and the inclusion of a random error term to separate noise from efficiency [27]. Another method involves data envelopment analysis (DEA), which is defined as a non-parametric mathematical approach that uses linear programming to construct a piece-wise frontier that serves as the foundation for evaluating relative efficiency for all firms [27]. Each method has its own advantages and disadvantages, but the appropriate choice depends on the research objectives, firms, and data availability. Due to the small sample size, we utilized DEA for the analysis. The primary advantage of this method is that neither the functional form of the production technology nor the testing of normality assumptions for the sample is required. Moreover, it can accommodate multiple inputs. In reference [28], the authors developed a DEA optimization tool with constant returns to scale (CRS), which became known as the CCR model. In this model, efficiency is explained in terms of revenue, technical, allocative, and costs/economics from the perspective of input and output orientations. However, the study focused on the three later types of efficiency from input orientation. The decision to assess technical, allocative, and economic efficiency was based on the research objectives and available data, while input orientation was chosen based on the fact that producers have control over the use of production inputs. In expressing the CCR model, let us assume that we have K inputs, M outputs, and N

broiler producers. This leads to the K × N input matrix for X and an M × N output matrix for Y. From this, efficiency is defined as the ratio of the weighted sum of outputs divided by the weighted sum of inputs. Using the notations introduced above:

$$u'y_i/q'x_i \tag{1}$$

where $u$ is an M × 1 vector of the output weights and $q$ is the K × 1 vector of the input weights. To find the optimal weights that allow us to measure efficiency, we solved the following mathematical programming problem:

$$\text{Max}_{u,q}\left(u'y_i/q'x_i\right) \tag{2}$$

subject to

$$u'y_i/q'x_j \leq 1, j = 1,2,\ldots N \tag{3}$$

$$u,q \geq 0 \tag{4}$$

This problem involves solving for $u$ and $q$, such that the measure of efficiency for the $i$-th firm is maximized subject to the constraints that all efficiency measures must be less than or equal to 1. The above formulation results in infinite solutions so it is necessary to reformulate the problem as follows:

$$\text{Max}_{\mu,v}\left(\mu'y_i\right) \tag{5}$$

subject to

$$v'x_i = 1 \tag{6}$$

$$\mu\prime y_j - v\prime x_j \leq 0, j = 1,2,\ldots N \tag{7}$$

$$\mu, v \geq 0 \tag{8}$$

In this formulation, we introduced an additional equation (Equation (5)); the notation was changed from $q$ and $u$ to $\mu$ and $v$. Another way of expressing the above model described from Equations (1) to (8) is:

$$\text{Min}_{\theta,\lambda}\ \theta \tag{9}$$

subject to

$$Y\lambda - y_i \geq 0 \tag{10}$$

$$\theta x_i - X\lambda \geq 0 \tag{11}$$

$$\lambda \geq 0 \tag{12}$$

where $\theta$ is a scalar and $\lambda$ is an N × 1 vector of constants. The estimated value of $\theta$ is the technical efficiency (TE) score for each of the N broiler producers. The estimate will satisfy the restriction $\theta \geq 1$ with a value $\theta = 1$ indicating a technically efficient producer. For economic efficiency, the model is:

$$\text{Min}_{x_i^*,\lambda}\ w'_i x'_i \tag{13}$$

subject to

$$Y\lambda - y_i \geq 0 \tag{14}$$

$$x_i^*\lambda - y_i \geq 0 \tag{15}$$

$$\lambda \geq 0 \tag{16}$$

where $w_i$ represents the input price vector for the $i$-th producer and $x_i^*$ is the cost-minimizing vector of input quantities for the $i$-th producer, given the input prices $w_i$ and the output

levels $y_i$. It follows that the EE of the *i*-th producer is then estimated as a ratio of the minimum cost to the observed cost, expressed as:

$$EE = \frac{w'_i x'_i}{w'_i x'_i} \tag{17}$$

$$AE_i = EE_i / TE_i \tag{18}$$

The determinants for the efficiency types were assessed using FRM, expressed as follows:

$$E_i = \zeta_0 + \sum_{n=1}^{4} \zeta_n Z_{ni} \tag{19}$$

where $E_i$ denotes the efficiency type while $\delta_0$, $\delta_m$, $\zeta_0$, and $\zeta_n$ are unknown parameters to be estimated using the fractional regression model. $Z_{mit}$, $Z_{ni}$, and $\zeta_n$ represent the covariates (mortality rate, heating cost, a dummy for female, and a dummy for Capricorn) that could explain variations in TE, AE, and EE, respectively. Reference [29] maintains that FRM is more appropriate compared to Tobit or OLS models. The efficiency scores were obtained using DEAP and the determinants using STATA 17. The results are discussed in the next section.

## 3. Results

### 3.1. Descriptive Summary for Types of Efficiencies

Table 2 presents the averages, standard deviations, minimums, and maximums for the outputs–inputs, and three efficiency types analyzed over the period of study. These results were obtained from an input-orientation perspective and dealt with adjusting inputs while keeping the output constant. As can be seen from the table, the results show an average TE score of 0.90%, indicating that the sampled smallholder broiler producers can adjust their current inputs without changing the output. This score (TE) had a range of 0.25% in Capricorn to 1.00% in both the Capricorn and Vhembe districts. Additionally, the input mix was incompatible with cost minimization; hence, the existence of the scope to improve EE by 28%. A similar finding was also observed for AE with a 20% potential for improvement on the input mix, taking into consideration their respective prices. EE scores ranged from 0.12% in Capricorn to 1.00% in the same district. The AE averaged 0.80% with a range from 0.11% to 1.00% in the Capricorn district. Overall, four smallholder broiler producers were economically efficient while only five were both technically and allocatively efficient. This implies that 95% of the sampled respondents were technically and allocatively inefficient. On the other hand, 96% were economically inefficient. Thus, the major finding that emerged was the effort to save on inputs or reduce waste. Details of the socioeconomic profile of the sampled respondents are presented in Table A1.

**Table 2.** Industry and regional productivity growth estimates.

| Estimates | TE (%) | AE (%) | EE (%) |
|:---:|:---:|:---:|:---:|
| Mean | 0.90 | 0.80 | 0.72 |
| Std | 0.14 | 0.11 | 0.16 |
| Min | 0.25 | 0.46 | 0.12 |
| Max | 1.00 | 1.00 | 1.00 |
| No. efficient producers | 5 | 5 | 4 |

### 3.2. Correlation for Output–Input Variables and Efficiency Types

Figure 2 presents the results of the correlation between the output–input variables and efficiency types. The purpose was to establish the relationship that existed among these variables. The value of a correlation coefficient can vary from minus one to plus one. Minus one indicates a perfect negative correlation, while plus one indicates a perfect positive correlation. A correlation of zero means implies no relationship between two

variables; here, it is a negative correlation between two variables, when increasing the value of one variable decreases the value of the other. The positive correlation between the two variables, on the other hand, exists when the value of one variable increases the value of the other variable. First, we found a strong positive correlation between the output and input, indicating that the output increases as the input increases. This finding is also confirmed by the scatter plots. However, it suggests that investing in smallholder broiler production is likely to yield the desired outcomes of the poultry sector's master plan in South Africa. It implies that great strides should be made toward proper implementation of the master plan to transform the industry, e.g., by promoting inclusive growth, defined as growth that is distributed fairly across society, creating opportunities for all. Such efforts will likely combat the growing imports of chickens, strengthen the local industry, and ensure that the country is self-sufficient in the long run.

In Figure 3, we examine the correlation by efficiency types. It is apparent from this figure that there exists a positive correlation between the three efficiency types. This indicates that a unit increase in one efficiency type leads to an increase in the other. However, the correlation coefficients vary significantly. For instance, a correlation coefficient of 0.33 was found between TE and AE, indicating a weak positive relationship between the two variables. However, the same variables had strong positive relationships with EE, indicating that transformation of smallholder broiler production can be achieved by placing more attention on economic efficiency. This finding implies that the analysis of efficiency in broiler production must not be limited to one single TE as is evident from the existing body of literature. Thus, a holistic approach is crucial for a comprehensive understanding and better policy insights. Nonetheless, there are two implications for the results on EE. Firstly, the deregulation of the agricultural sector does not improve competition as smallholder broiler farmers are concerned. This suggests that state intervention, as opposed to market forces alone, should also be considered to remedy anti-competitive behaviors. Secondly, the South African Poultry Association should strengthen its unit that deals with smallholder broiler development.

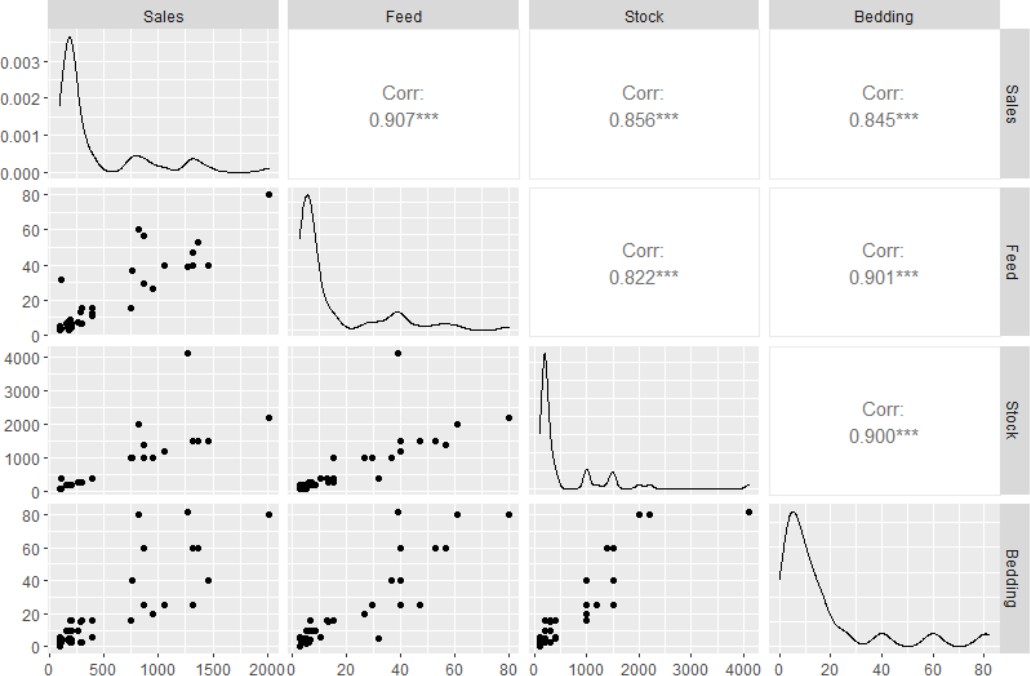

**Figure 2.** Output and input correlations along with their density plots and scatter plots for the entire sample. *** $p < 0.001$.

Previous findings were limited to the smallholder sectoral level. For practical application, we delved into the specifics of broiler producer efficiency by district (see Figure 3).

Given the recent developments in using the district model as an implementation tool for government programs, this analysis is critical.

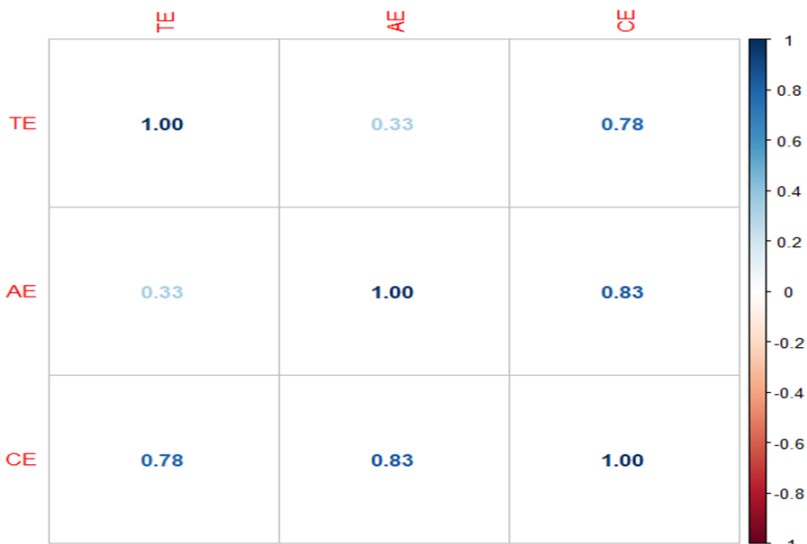

**Figure 3.** Correlations for efficiency types.

*3.3. District-Wise Efficiency of Broiler Producers*

The South African government has adopted the district development model (DDM) as one of the primary tools to improve service delivery, in view of achieving the goals of the National Development Plan Vision 2030. Briefly, the DDM consists of a process in which all three spheres of government undertake joint and collaborative planning at local, district, and metropolitan levels, resulting in a single, strategically-focused plan ('one plan') for each of the country's 44 districts and 8 metropolitan geographic areas, with the district serving as the 'landing strip'. Against this background, an analysis of efficiency at the district level was deemed important to uncover the practical value of the study. Figure 4 depicts the technical efficiency (TE) on the *y*-axis and the allocative efficiency (AE) on the *x*-axis, with various districts in the legend. The legend also indicates that the larger the bubble, the higher the EE or CE. The most interesting of this figure is that only the Capricorn district had broiler producers who were technically, allocatively, and economically efficient. These farmers were estimated at 6%, translated as 4, implying that more effort is required to uplift the majority of the sampled respondents towards the desired levels for different types of efficiency. Only two producers were technically efficient in Vhembe and no attainment of allocative and cost efficiencies was found. The efficiency scores in DRKK were far below the desired outcomes for each efficiency type, indicating that more intervention is required in this particular district. On average, Capricorn had 0.92% in TE with AE and EE at 0.83% and 0.77%. The average efficiency scores in DRKK were 0.80% (TE), 0.79% (AE), and 0.64% (EE), while the Vhembe district recorded 0.91% (TE), 0.77% (AE), and 0.70% (EE). Increased access to extension services, credit, and competition reforms could all be part of the intervention. Overall, the variation of scores (by efficiency types) indicate that the districts are different in terms of resource endowments, technology, and climate and, thus, necessitate the formulations of district-specific policies.

*3.4. Economic Efficiency by Gender of Broiler Producers*

The study also investigated the efficiencies of sampled respondents by gender distinction. This analysis was conducted in view of the growing interest in sustainable agriculture through greater levels of inclusion of women. We discovered that male broiler producers operated on the upper bounds of TE, at an average of 0.91%. Female broiler producers operated on the upper bounds of AE, at an average of 0.81%. This result suggests that males have a greater ability to exploit available inputs toward broiler production compared

to females. However, females exhibited more potential to use inputs in optimal proportions, given their respective prices and production technology. In terms of EE, both males and females had the same average score of 0.72%. This suggests that both females and males are likely to show the same economic performance (given equal opportunities). Further analysis (by gender, per district) showed that both males and females had similar CE scores in the Capricorn district (see Figure 5). However, there were clear differences in DRKK, with females operating on upper bounds compared to males (and vice versa) in the Vhembe district. A possible explanation for this finding is inequitable access to factors of production among males and females. The average efficiency scores for females in the Capricorn district were 0.92% (TE), 0.81% (AE), and 0.76% (EE). For males, the average efficiency scores were O.99% (both TE and AE) and 0.98% (EE) in the same district. In DRKK, female broiler producers had average efficiency scores of 0.78% (TE), 0.79% (AE), and 0.69% (EE) with males at 0.89% (TE), 0.78% (AE), and 0.69% (EE). For males in the Vhembe district, the scores were 0.90% (TE), 0.77% (AE), and 0.70%, while for females, the scores were 0.94% (TE), 0.77% (AE), and 0.73% (EE). Overall, these findings suggest variations in male and female performances in smallholder broiler production. Thus, policy interventions should be customized toward specific needs by gender distinction. In this way, sustainable broiler production and inclusive growth are more likely to be realized.

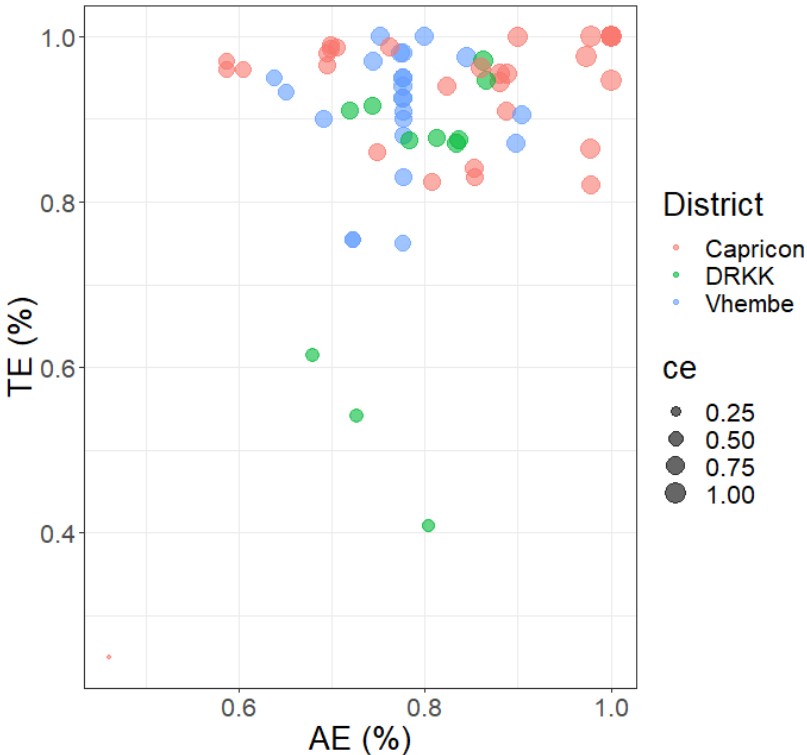

**Figure 4.** Evaluation of each type of efficiency by district.

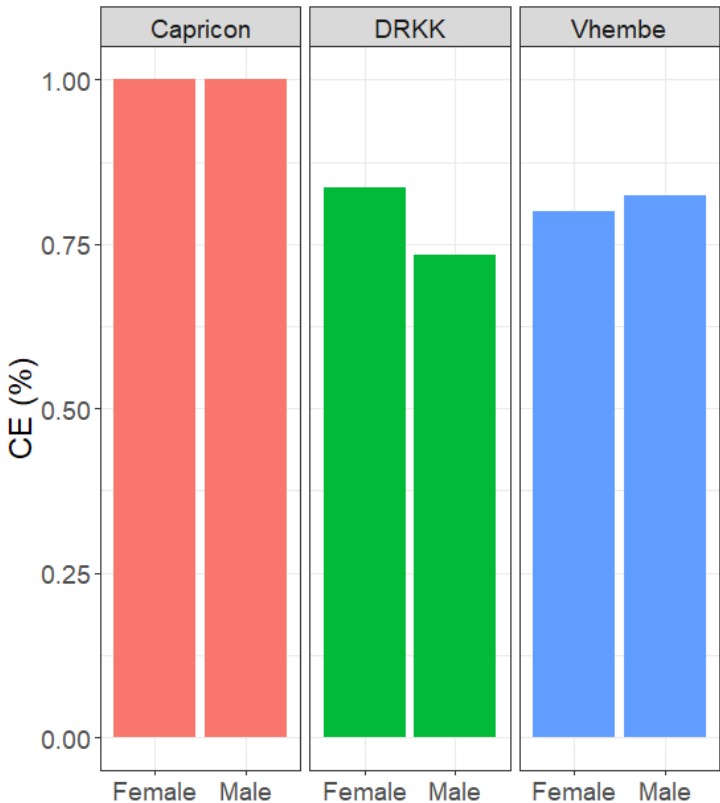

**Figure 5.** Evaluation of efficiency by type of gender for each district.

### 3.5. Sources for Each Type of Efficiency

This section is designed to address the second research question of the study. Three separate fraction regression models were estimated to assess factors influencing each efficiency type and the results are reported in Table 3. The results show that most of the explanatory variables were significant at the 5% level. The variable mortality rate had a negative association, indicating that more death rates in reared broiler birds tend to lead to technical and economic inefficiencies. The coefficient of variable heating cost had a negative sign, implying that more electricity expenses are associated with technical and economic inefficiencies. The negative signs of the effects of gender on the inefficiency values imply that the TE, AE, and CE levels of smallholder broiler producers tended to be less among males. This finding has implications for gender inclusion in smallholder broiler production in South Africa. The coefficients of the district dummy variables show positive signs, indicating that the levels of AE and CE tended to be high in the Capricorn district compared to other districts.

**Table 3.** Efficiency determinants from the fractional regression model.

| Estimates | TE | | AE | | CE | |
|---|---|---|---|---|---|---|
| | Coef. | SE | Coef. | SE | Coef. | SE |
| Intercept | 3.87 *** | 0.56 | 1.03 *** | 0.34 | 1.95 *** | 0.36 |
| Mortality rate (%) | −0.49 *** | 0.07 | 0.11 *** | 0.05 | −0.09 | 0.05 |
| Heating cost (R) | −0.23 *** | 0.01 | −0.01 | 0.01 | −0.02 *** | 0.00 |
| Dummy males | −0.38 *** | 0.17 | −0.14 | 0.12 | −0.34 *** | 0.12 |
| Dummy Capricorn | 0.26 | 0.17 | 0.32 ** | 0.14 | 0.44 *** | 0.13 |
| Pseudo R square | 0.13 | | 0.01 | | 0.03 | |
| Wald chi-square | 53.50 *** | | 8.37 * | | 30.01 *** | |
| Log pseudo-likelihood | −18.13 | | −30.91 | | −35.62 | |

Note: * $p < 0.05$, ** $p < 0.01$, *** $p < 0.001$

## 4. Discussion

Concerning the first research question, the study found an average TE of 0.90%. This is slightly lower (by 3%) compared to the estimate by [11] in Thailand. It is also lower compared to what was reported in [30], where the authors reported average TE scores of 0.88% and 0.83% under CRS and variable return to scale (VRS) models for 120 Nigerian broiler producers. From the same sample sizes of broiler producers in Nigeria, a study by [31] conducted through the SFA method reported an average TE score of 0.78% (for both contract and non-contract producers). Thus, had we used SFA instead of DEA, our study would have yielded different estimates. A recent study by [32], utilizing SFA, reported a higher estimate of the average TE compared to our study. However, it was earlier indicated (in the introduction) that a small sample size was the limitation to the exploration of SFA. Unfortunately, we only found one published study [24] exploring broiler efficiency in South Africa. However, it does not shed light on the estimates of TE. Despite this, the most common feature of our study (and existing literature) is that of identifying the scope to improve the input use without changing the level of the output.

Our study findings also reveal an average AE score of 0.80%. This result is contrary to [33], who reported allocative efficiency for 96 broiler producers in Nigeria. However, it corroborates the study in [11], where the authors reported an average AE estimate of 0.88%. Furthermore, our findings are also supported by [34], where the authors discovered a mean allocative efficiency score of 0.84% in 120 broiler farms in Pakistan using the Cobb–Douglass SFA. However, they remained lower than that of [35]; the authors of that study showed average AEs of 0.70% (CRS) and 0.73% (VRS) for 59 broiler farmers in Iran. Moreover, they corroborated a recent study by [30], where the authors reported on the lack of cost minimization among broiler producers in Nigeria. This implies that smallholder broiler producers in South Africa need to improve on the optimum use of inputs given their costs. We further found an average EE of 0.72%. This differs from [36], possibly due to differences in the sample sizes and methods used. Nonetheless, we consistently observed that EE estimates turned out to be low compared to AE and TE. The available literature studies [11,35] confirm this observation.

The estimates for the three efficiency types differed by district. Most studies that investigated various types of broiler efficiencies have been conducted in Nigeria. These include [15,30,33]; their estimates vary by state. This finding suggests that district-specific interventions are also too important to be ignored. This variation was also noticed when analyzing the three efficiency types by gender distinction; thus, supporting [37,38]. We could not find literature to support our findings on the efficiency determinants due to the fact that most of the determinant variables have not been explored in the past. According to [39], this inefficiency could be addressed through the provision of subsidized financing, increased extension services to enhance production, and decreased production costs. Such interventions are supported by [40], who proposed focusing on the personal characteristics of farmers and improving their farming profiles.

## 5. Conclusions

Returning to the first research question posed at the beginning of the study, it is now possible to conclude that, on average, the sample as a whole exhibited technical, allocative, and economic inefficiencies. This implies the potential for smallholder farmers to reduce input use and costs without adjusting the output. However, this potential varies per efficiency type, but largely suggests that the transformation of smallholder broiler farmers could be achieved by addressing cost inefficiency. Another finding supporting the transformation is the strong positive correlation between outputs and inputs, suggesting that input intensification by these farmers is more likely to bring about the desired outcomes of the South African Poultry master plan. District-wise, Capricorn was the only district with technical, allocative, and cost-efficient broiler producers. Only two producers were technically efficient in the Vhembe district. No evidence of technical, allocative, or cost

efficiencies were found in DRKK. Another important finding related to the first research question involved variation in each efficiency type by gender.

Regarding the question of determinants for each efficiency type, the study found that more death rates of reared broiler birds was associated with technical and economic inefficiencies. Moreover, the high mortality rates could lead to optimum use of resources to rear the available stock (hence, the positive association between the mortality rate and allocative efficiency). As expected, high heating costs were associated with technical and economic inefficiencies. One of the more significant findings to emerge from this study is that a unit increase of male smallholder broiler producers was associated with technical and economic inefficiencies. We also found that allocative and economic efficiencies were significantly determined by location, with Capricorn more efficient compared to the other districts. Taken together, these findings suggest that 'efficiency research' on smallholder broiler production is crucial to promote the transformation of the country's broiler industry.

Despite the small sample size, the study has provided deeper insight into the efficiencies of smallholder broiler producers by incorporating allocative, economic, and cost efficiencies, which have received very little attention (regarding South Africa) in the current body of knowledge. This new understanding should help improve resource allocations and promote inclusive growth. As a result, several policy recommendations have arisen from the study. First, training and education through the provision of agricultural extension services are crucial to improving technical efficiency. Improved access to information (concerning inputs and their respective prices) can also assist farmers to improve optimum resource allocations. Gender-specific strategies are needed to promote inclusive growth since females perform better (in terms of efficiency) than their male counterparts. Lastly, the district development model should be adopted as a tool to implement these policy recommendations.

**Author Contributions:** Conceptualization, L.W.M. and N.B.N.; Literature, E.N.; methodology, L.W.M.; Data curation, L.M. and N.B.N.; formal analysis, L.W.M.; review and editing, S.N.; Sourcing publication fee, S.N. and N.B.N. All authors have read and agreed to the published version of the manuscript. have read and agreed to the published version of the manuscript.

**Funding:** This research received publication fee from the University of South Africa and National Agricultural Marketing Council through the effort of N.B.N and S.N.

**Data Availability Statement:** Data are available upon request from the lead author, LW Myeki.

**Conflicts of Interest:** The authors declare no conflict of interest.

**Appendix A**

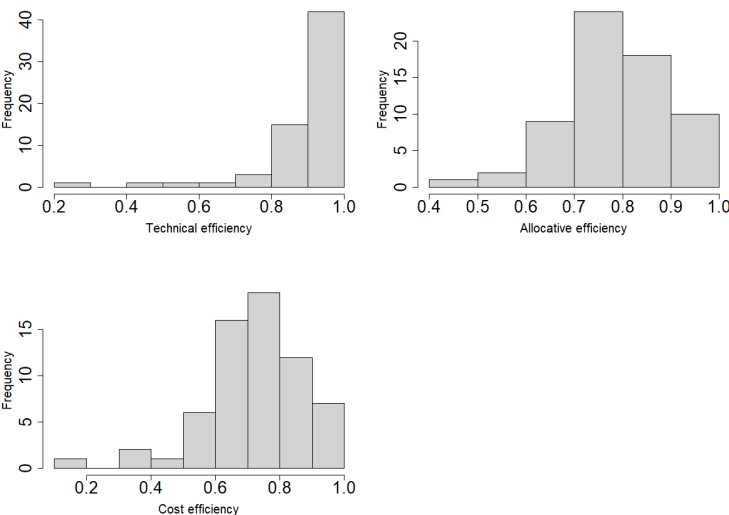

**Figure A1.** Efficiency distribution for sampled respondents.

**Table A1.** Socioeconomic profiles of the sampled respondents.

| Variable | Category | Districts | | | Total |
|---|---|---|---|---|---|
| | | DRKK | Capricorn | Vhembe | |
| Gender (n%) | Female | 9 (82) | 30 (94) | 4 (17) | 43 (65) |
| | Male | 2 (18) | 2 (6) | 19 (83) | 23 (35) |
| Age (mean year) | Adult | 54 | 58 | 52 | 55 |
| | Youth | 31 | 32 | 39 | 33 |
| Input cost (R) | Mean | 28,914 | 7902 | 9364 | 12,039 |
| | Std | 27,159 | 3344 | 9325 | 14,593 |
| | Min | 1680 | 1770 | 955 | 955 |
| | Max | 69,484 | 17,107 | 49,720 | 69,484 |
| Income (R) | Mean | 89,300 | 14,821 | 20,109 | 29,672 |
| | Std | 26,447 | 5620 | 19,387 | 32,014 |
| | Min | 49,140 | 6570 | 7040 | 6570 |
| | Max | 131,130 | 26,045 | 81,975 | 131,130 |
| Profit (R) | Mean | 60,387 | 6919 | 9871 | 17,170 |
| | Std | 41,085 | 4299 | 14,824 | 27,364 |
| | Min | −15,479 | 98 | −955 | −15,479 |
| | Max | 116,496 | 18,971 | 51,810 | 116,496 |

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
