# Peer review of "Estimation of Technical, Allocative, and Economic Efficiencies for Smallholder Broiler Producers in South Africa"

_agriculture, doi:10.3390/agriculture12101601_

Round 1
Reviewer 1 Report
The manuscript entitled “Estimation of technical, allocative and economic efficiencies for smallholder broiler producers in South Africa” is an interesting study in which authors attempt to estimate three types of efficiencies of broiler producers in South Africa. The data envelopment analysis approach has been used to estimate the efficiencies. Using cross-sectional data, the authors also attempt to determine the source of efficiency. The paper is well-written and well-organized. The findings of the study have potential policy implications. Furthermore, the article is relevant to the scope of the journal. However, some revisions are required to improve the quality of the manuscript. Below, I am making my comments on several sections of the manuscript:
Introduction
The introduction section is mostly well-written. The authors have first introduced the problem of broiler production in terms of widening gap between production and consumption in South Africa. Then, the authors provide a brief review of existing studies on broiler efficiency and methodologies used to estimate technical efficiencies. Finally, they state the research gap and objectives. There are some minor issues to address here though:
What is 47.96 billion on Line 20. Please mention currency unit.
Line 21 starts with “their meat”. Please revise. The pronoun does not specify what we are talking about here.
Methodology
Please provide a Study Area map for the better understanding of the region in which the study is being conducted.
What is DALRRD production scheme on Line 99?
How can the authors explain the choice of ‘inputs’ in the model? There seems to be too few input variables. For example, why did the authors not include heating cost as one of the variables in the DEA model? How about management cost, labor cost, and other significant costs?
The same issue applies to the choice of variables for the regression model. In the Introduction section, the authors state that they have used ‘fractional regression model’ to estimate the determinants of efficiency. However, no details about the regression model have been provided in the methodology.
Results
Please provide a Table including the socioeconomic profile of the sample respondents/farms in order to understand your sample.
Since the results are rather simplistic, the authors need to make an effort to dig deeper and provide further analysis. For example, instead of providing the mean of efficiencies, please provide a Table of efficiency classes (frequency distribution) in order to better understand the distribution of efficiencies. Please refer to the following article:
Razzaq et al. (2019). Can the informal groundwater markets improve water use efficiency and equity? Evidence from a semi-arid region of Pakistan. Science of the Total Environment, 666, 849-857.
You may also cite the above article in the literature review as it uses a truncated regression model to estimate the sources of inefficiency. Earlier, the authors have stated that their article departs from the tradition of using ‘tobit’ model as determinants of (in)efficiency.
The authors have not used any statistical tests to determine whether the efficiency differentials are significant across groups.
Please consider changing the heading 3.3 to ‘District-wise efficiency of broiler producers’.
Author Response
Dear Reviewer,
We sincerely appreciate the constructive inputs that have assisted us to improve the quality of the paper. Attached please find our responses to each comment/input.
Cheers
Lindikaya Myeki

Reviewer 2 Report
The manuscript draft is devoted to an interesting problem that touches on the efficiency of broiler producers, using a two-stage Data Envelopment Analysis method with input orientation. The proposed approach is logical, results are clear. However, I have the following remarks:
1. The authors mentioned that the prevailing economic condition as a result of COVID-19, climate change and the Russia-Ukraine conflict has led to renewed global interest in agricultural sector efficiency. The author should describe in detail the real situation. It would be better to add some graphs and numbers.
2. In the methodology section, the authors describe the data, sample, and estimation procedure. The authors should describe the reasoning for choosing DEA. Also, it will be good to explain the criteria used in selecting an approach to the research. The authors should list potential weaknesses in methodology and present evidence supporting their choice.
3. The authors should describe the policy recommendations in more detail. This will add value to the paper.
4. The conclusion is not explained properly. The conclusion section should be extended using: the results of the research, a discussion of related research, and a comparison between the authors’ results and initial hypothesis.
5. The references may need to be updated.
Author Response

(The authors gave the same response as above.)

Round 2
Reviewer 1 Report
Since the authors have made the necessary improvements suggested in the first round of revisions, the paper can now be accepted in its present form.